# A Study on the Vortex Induced Vibration of a Cylindrical Structure with Surface Bulges

Haoyuan Xu [ID], Jie Wang, Zhiqing Li [ID], Kaihua Liu [ID], Jiawei Yu and Bo Zhou *

State Key Laboratory of Structural Analysis for Industrial Equipment, School of Naval Architecture Engineering, Dalian University of Technology, Dalian 116024, China
* Correspondence: bozhou@dlut.edu.cn

**Abstract:** Inspired by the cactus in nature, a cactus-like cross-sectional structure was proposed to achieve the VIV suppression. The VIV of the elastically mounted cylinder was realized based on the ANSYS Fluent and User Defined Function (UDF). The dynamic motion of the cylinder was solved by the single-step time integration algorithms Newmark-$\beta$ method. The in-house code was first validated by studying the 2DOF VIV of a circular cylinder with small mass ratio over the range $U^* = 2 \sim 13$, and the results agree well with the published literature. Then, the performance of surface bulge on VIV suppression was studied and four different coverage ratios (CR) were considered, i.e., 0%, 20%, 33%, and 40%. The VIV of a bulged cylinder can be effectively suppressed. CR20 performs the best in VIV suppression and the suppression efficiency in streamwise and transverse direction are 44.6% and 63.1%, respectively. The mechanism of surface bulge on the VIV suppression is the shift of separation point of the shear layer and vortices form between the surface bulges.

**Keywords:** offshore riser; vortex-induced vibration; surface bulge; UDF

## 1. Introduction

The cylindrical structures are commonly applied in offshore engineering, such as platform legs, offshore risers, seabed pipelines, etc. Alternative vortex shedding is typically generated at both sides of the cylinder when flow is past the cylinder. The shedding of the vortices can lead to large fluctuating forces on the cylinder, which can induce the vibration of the cylinder. When the vortex shedding frequency is close to the natural frequency of the structure, it will lead to the synchronization of the structure (also known as lock-in) and failure of the structure in some extreme cases. Up to now, extensive studies have been carried out to uncover the mechanism behind VIV generation (Williamson and Govardhan [1,2], Pan [3], Gabbai [4]).

The VIV suppression of the riser is an important research topic in the field of offshore and engineering. The suppression of VIV can be realized from two aspects: structure and flow field. The VIV of the riser can be constrained by changing the structure properties, such as top tension, structural stiffness and damping. The flow field is mainly altered by modifying the vortex shedding, which can be classified into active control and passive control [5]. The active control method suppresses vortex shedding by injecting energy into the flow field, which introduces external disturbances in the flow field. The passive control method changes the flow field in the vicinity of the structure by modifying the cross-section shape or adding additional structures. These structures can effectively modify the shear layer separation and weaken the interaction of the shear layer. Due to its simple operation, high efficiency and low cost, the passive control is widely used in the field of ocean engineering.

Passive control devices can be generally classified into three categories based on their functions: (i) modification of the shear layer separation, including helical strakes [6], surface roughness [7], spoilers [8], traveling wave walls [9], etc. The modification of

the separated shear layer further changes the vortex shedding mode; (ii) wake stability, including fairing [10] and splitter plate [11,12], etc. These structures shift the shear layer separation point and weaken the inter-action of the shear layer, which stabilizes the wake structures and affects the vortex shedding mode; and (iii) Suppression of vortex formation and separation, including control rod [13,14], groove [15], etc. Helical strakes and fairings are extensively studied among all the passive control devices. Helical strakes can effectively suppression the VIV of structure. However, a dramatic drag increase will occur owing to the pressure drop right behind the structure. The fairing device can diminish the VIV response of the structure and reduce the drag force acting on the cylinder, while complex installation and high cost make it difficult to be widely applied.

Regarding the above problems, many scholars, inspired by the cactus in nature (see Figure 1a), study the VIV suppression of a cactus-like cross-sectional structure, also known as bulged surface. Marcollo et al. [15] studied the VIV of a cylinder with cactus-like cross-section experimentally and proposed a longitudinal groove control method. The results indicate that the performance of the longitudinal grooved cylinder is related to the shape and number of grooves. Talley et al. [16] conducted an experimental study of cactus-like cylinders with different grooved depths, and the results show that the drag forces acting on the cylinder decrease as the grooved depth increases for Reynolds number in the range $2 \times 10^4 \sim 2 \times 10^5$. Wang et al. [17] numerically studied flow past a cylinder with a fixed surface bulge. The forces acting on the cylinder and Strouhal number are smaller than that of a smooth cylinder. Qian et al. [18] carried out a three-dimensional numerical simulation of flow past a cactus-like cylinder and a circular cylinder. It was found that the cactus-shaped cylinder had a good performance on drag reduction. Zhou et al. [19] experimentally studied the grooved and dimpled cylinders with a fixed groove depth of 0.05 D. A reduction on mean drag $C_D^M$ and root mean square lift $Cl_{r.m.s}$ coefficients was achieved over the range of $7.4 \times 10^3 \le \text{Re} \le 8 \times 10^4$. The visualization of the flow field indicates that the groove and dimple structures diminish the magnitude of vortex shedding. Wang et al. [20,21] investigated the effect of bulge height and number of ribs on VIV suppression, and the results showed that the surface bulges can effectively suppress structural vibration.

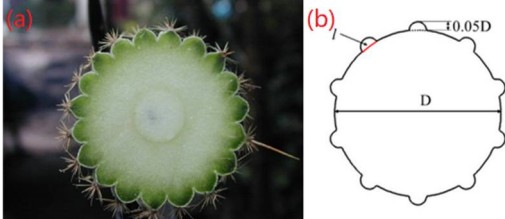

**Figure 1.** (**a**) Cross-section of the cactus and (**b**) the sketch of a cylinder with surface bulges.

In light of the published literature, the study of VIV suppression is limited to a cylinder with large and medium mass ratios $m^*$ ($m^* = m/m_d = 4m/(\rho\pi D^2 L)$, $m$ is the mass of the structure; $m_d$ is the mass of displaced fluid; $\rho$ is the fluid density; $D$ is the diameter of the structure; $L$ is the length of the structure). The research on the VIV of a cylinder attached with a surface bulge at a small mass ratio is relatively unexplored. In this paper, the influence of the coverage ratio on the VIV of a circular cylinder attached with surface bulges is systematically studied. The coverage ratio is defined as $CR = l/\pi D$, l is the length of the bulge as depicted in Figure 1b, and the height of each bulge is fixed at 0.05 D. The coverage ratio varies by adjusting the number of bulges. Four coverage ratios, i.e., 0%, 20%, 33%, and 40% were considered in this paper, as shown in Figure 2. In the proceeding sections of this paper, the coverage ratios are denoted as CR0, CR20, CR33, and CR40, respectively.

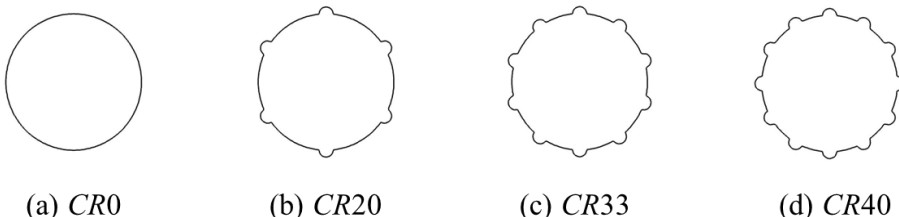

(a) *CR0*　　　(b) *CR20*　　　(c) *CR33*　　　(d) *CR40*

**Figure 2.** Schematic diagram of a cylinder with different coverage ratios.

This paper is organized as follows: Section II describes the numerical method and is followed by a model validation of the in-house code with the classical VIV experiment [22] in Section III. The influence of coverage ratio on the VIV of a cylinder attached with surface bulges are presented in Section IV, including vibration amplitude, vortex shedding frequency, wake modes, mechanism of VIV suppression, etc.

## 2. Numerical Method

### 2.1. Governing Equations

The governing equations for two-dimensional unsteady incompressible flow are the continuity equation and Navier Stokes equations:

$$\frac{\partial u_i}{\partial x_i} = 0 \tag{1}$$

$$\frac{\partial u_i}{\partial t} + u_j \frac{\partial u_i}{\partial x_j} = -\frac{1}{\rho}\frac{\partial p}{\partial x_i} + \frac{\partial}{\partial x_j}\left(v\frac{\partial u_i}{\partial x_j}\right) - \frac{\partial \overline{u_i' u_j'}}{\partial x_j} \tag{2}$$

where $u_i$ is the velocity component in the $x_i$ direction; $\rho$, $p$, and $v$ are the fluid density, pressure, kinematic viscosity; $\overline{u_i' u_j'}$ is Reynolds stress tensor.

In this paper, the governing equation is discretized by the finite volume method (FVM); the pressure-velocity coupling equation is solved by SIMPEC algorithm; the PRESTO is adopted for pressure discretization; the QUICK scheme is employed for momentum term; the turbulent kinetic energy and dissipation terms are solved by the second-order upwind scheme and first-order upwind scheme, respectively; and the transient formula is solved by the first order implicit method.

### 2.2. SSTk-ω. Turbulence Model

The turbulence kinetic energy $k$ and specific dissipation rate $\omega$ are given as:

$$\frac{\partial(\rho k)}{\partial t} + \frac{\partial(\rho u_j k)}{\partial x_j} = \tau_{ij}\frac{\partial u_i}{\partial x_j} - \beta^* \rho k\omega + \frac{\partial}{\partial x_j}\left[(\mu + \sigma_k \mu_t)\frac{\partial k}{\partial x_j}\right] \tag{3}$$

$$\frac{\partial(\rho \omega)}{\partial t} + \frac{\partial(\rho u_j \omega)}{\partial x_j} = \frac{\gamma}{v_t}\tau_{ij}\frac{\partial u_i}{\partial x_j} - \beta\rho\omega^2 + \frac{\partial}{\partial x_j}\left[(\mu + \sigma_\omega \mu_t)\frac{\partial \omega}{\partial x_j}\right] + 2\rho(1 - F_1)\sigma_{\omega 2}\frac{1}{\omega}\frac{\partial k}{\partial x_j}\frac{\partial \omega}{\partial x_j} \tag{4}$$

where $u_i$ and $u_j$ are the velocity components in the $x$ and $y$ direction, respectively; $k$ is the turbulent kinetic energy; $\omega$ is the specific dissipation rate; $\tau_{ij}$ is Reynolds stress; $\mu$ is the dynamic viscosity; $v_t = \mu_t/\rho$ is kinematic eddy viscosity; $\mu_t$ is eddy viscosity.

$$\mu_t = \frac{\alpha_1 \rho k}{\max(\alpha_1 \omega, \|\mathbf{\Omega}\| F_2)} \tag{5}$$

where $\|\mathbf{\Omega}\| = \sqrt{2\Omega_{ij}\Omega_{ij}}$, the vorticity magnitude with $\Omega_{ij} = \frac{1}{2}\left(\frac{\partial u_i}{\partial x_j} - \frac{\partial u_j}{\partial x_i}\right)$.

Auxiliary Relations

$$F_1 = \tanh\left\{\left\{\min\left[\max\left(\frac{\sqrt{k}}{\beta^* \omega y^*}, \frac{500\nu}{y^{*2}\omega}\right), \frac{4\sigma_{\omega 2}\rho k}{CD_{k\omega}y^{*2}}\right]\right\}^4\right\} \tag{6}$$

$$CD_{k\omega} = \max\left(2\rho\sigma_{\omega 2}\frac{1}{\omega}\frac{\partial k}{\partial x_j}\frac{\partial \omega}{\partial x_j}, 10^{-20}\right) \tag{7}$$

$$F_2 = \tanh\left\{\left[\max\left(\frac{2\sqrt{k}}{\beta^* \omega y^*}, \frac{500\nu}{y^{*2}\omega}\right)\right]^2\right\} \tag{8}$$

where $y^*$ is the distance from the field point to the nearest wall.

The blending function $F_1$ takes the value 1 in the inner parts of the boundary layer ($y^* < \delta$, where $\delta$ is the thickness of the boundary layer) and 0 in the free-stream ($y^* > \delta$), completing the switch from $k - \omega$ model to $k - \varepsilon$ model. Accordingly, the coefficients in the transport equation can be expressed by $F_1$:

$$\phi = F_1\phi_1 + (1 - F_1)\phi_2 \tag{9}$$

where the value of each closure coefficients are as follows: $\alpha_1 = 0.31; \beta^* = 0.09; \kappa = 0.41; \sigma_{\omega 1} = 0.5; \sigma_{\omega 2} = 0.856; \sigma_{k1} = 0.85; \sigma_{k2} = 1.0; \beta_1 = 0.075; \beta_2 = 0.083$.

*2.3. Kinematic Equations*

The dynamic motion of an elastically mounted cylinder can be expressed by:

$$m\ddot{x} + C\dot{x} + Kx = F_D(t) \tag{10}$$

$$m\ddot{y} + C\dot{y} + Ky = F_L(t) \tag{11}$$

$$m = m^* \pi \rho D^2 / 4 \tag{12}$$

$$M = (C_A + m^*)\pi \rho D^2 / 4 \tag{13}$$

$$C = 2\sqrt{KM}\zeta \tag{14}$$

$$\omega_0 = 2\pi\sqrt{K/M} \tag{15}$$

where $K$ is stiffness; $C$ is damping; $F_D(t)$ and $F_L(t)$ are drag and lift force, respectively. $\rho$ is the density of water. $C_A$ is the added mass coefficient, taken as $C_A = 1$; $\omega_0$ is the circular frequency of the cylinder; $\zeta$ is the damping ratio.

In this paper, the weak-coupling method is employed to simulate the VIV response of cylindrical structures. The kinematic equations are solved by Newmark-$\beta$ method [23]. The derivation of the structure velocity and displacement in the transverse direction is similar to that in streamwise direction, so only the derivation in the streamwise direction is presented herein.

$$\widetilde{F_D}(t + \Delta t) = F_D(t + \Delta t) + m\left(\alpha_0 x(t) + \alpha_2 \dot{x}(t) + \alpha_3 \ddot{x}(t)\right) + c\left(\alpha_1 x(t) + \alpha_4 \dot{x}(t) + \alpha_5 \ddot{x}(t)\right) \tag{16}$$

$$\widetilde{K} = K + \alpha_0 m + \alpha_1 c \tag{17}$$

$$x(t + \Delta t) = \widetilde{F_D}(t + \Delta t)/\widetilde{K} \tag{18}$$

$$\ddot{x}(t + \Delta t) = \alpha_0(x(t + \Delta t) - x(t)) - \alpha_2 \dot{x}(t) - \alpha_3 \ddot{x}(t) \tag{19}$$

$$\dot{x}(t + \Delta t) = \dot{x}(t) + \alpha_6 \ddot{x}(t) + \alpha_7 \ddot{x}(t + \Delta t) \tag{20}$$

where $\widetilde{K}$ is equivalent stiffness; $\widetilde{F_D}$ is equivalent load; $\alpha_0 = 1/\beta\Delta t^2$; $\alpha_1 = \gamma/\beta\Delta t$; $\alpha_2 = 1/\beta\Delta t$; $\alpha_3 = (1-2\beta)/2\beta$; $\alpha_4 = (\gamma-\beta)/\beta$; $\alpha_5 = (\Delta t/2)((\gamma-2\beta)/\beta)$; $\alpha_6 = \Delta t(1-\gamma)$; $\alpha_7 = \gamma\Delta t$; $\Delta t$ is time step, chosen as 0.005 s. This method is unconditionally stable when $\alpha \geq 0.5$, $\beta \geq (0.5+\alpha)^2/4$, thus $\alpha$ and $\beta$ are chosen as $\alpha = 0.5$, $\beta = 0.25$.

The solving procedure can be summarized as follows (See Figure 3):

Step 1:  Extract the forces in both streamwise and transverse direction by the macro Compute_Force_And_Moment;

Step 2:  Substitute the force in the right hand of Equation (16);

Step 3:  Calculate the displacement, acceleration, and velocity of the cylinder by Equations (18)–(20);

Step 4:  Assign the displacement and velocity to the centroid of the cylinder by the macro DEFINE_CG_MOTION.

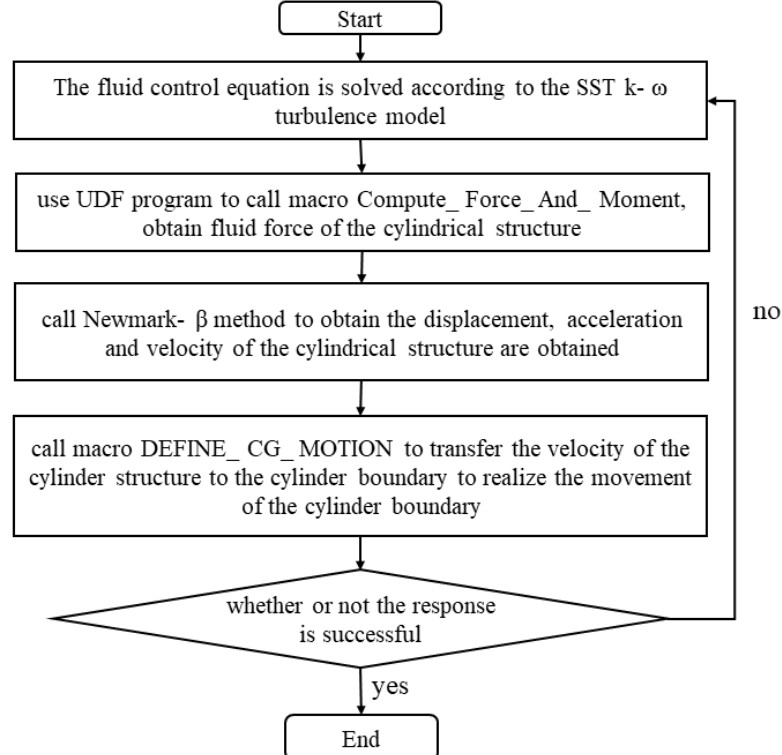

**Figure 3.** Program diagram for calculation.

## 3. Computational Model

### 3.1. Computational Domain

The sketch of the computational domain is shown in Figure 4. The origin of the Cartesian coordinate is located at the center of the cylinder. The longitudinal and transverse length of the computational domain are 35 D and 20 D, respectively. The inlet is located at 10 D upstream and the outlet is assigned at 25 D downstream. Two lateral boundaries are symmetric about the $x$ axial, which is 10 D away from the origin. The whole computational domain is divided into two layers, i.e., the attached layer (marked in gray) and the deformation layer. The attached layer is defined by the region around the cylinder with an outer diameter of 2 D. The attached layer moves with the cylinder, and the mesh inside this region remains unchangeable during the whole calculation. The meshes in the deformation layer are updated based on the conservation law of dynamic meshes.

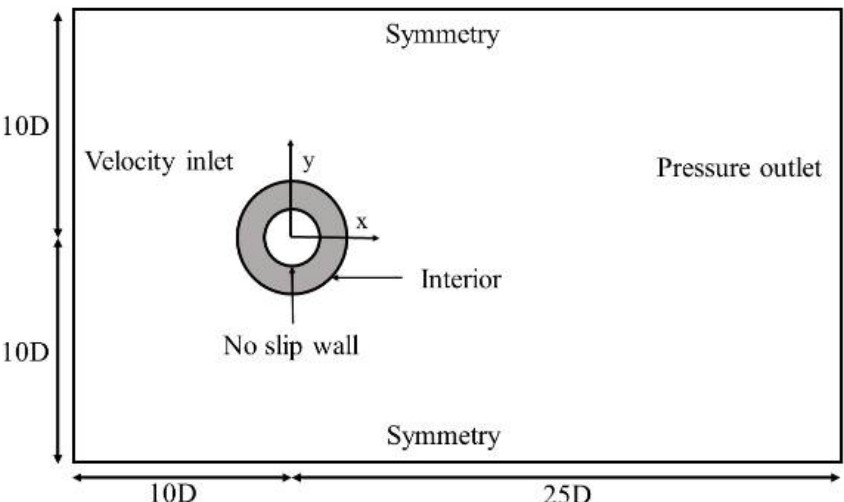

**Figure 4.** Sketch of the computational domain.

### 3.2. Boundary Conditions

The Dirichlet and Neumann boundary condition were specified at the inlet and outlet, respectively; The two lateral boundaries were assigned with symmetry boundary condition, and the no-slip boundary condition was employed on the cylinder surface.

### 3.3. Mesh Setup

Hybrid meshes were employed for the discretization of the computational domain and convergence of the simulation (See Figure 5). Fine structural meshes were used in the attached zone. The height of the first layer is located at $\Delta y = 0.002$ D away from the cylinder, with a stretching rate 1.2, and the circumference of the cylinder was uniformly divided into 320 grids. The unstructured meshes were applied in the defamation zone, and the base size of the meshes in this region was chosen as 0.2 D.

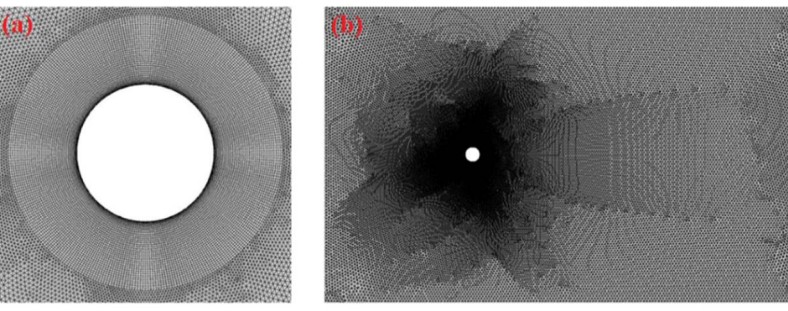

**Figure 5.** Discretization of the computational domain (**a**) close-up of the meshes near the cylinder; (**b**) Mesh division of the computational domain.

### 3.4. Mesh Independence Study

In this paper, the turbulence intensity at the inlet boundary was set as 9%, and the turbulent length scale was taken as 0.04 D (where the turbulence intensity is defined as the ratio of the root mean square of the fluctuation velocity to the mean velocity; the turbulence length scale is related to the size of the eddies), referred to Han [24]. The parameters listed in Table 1 were consistent with the classical VIV experiment conducted by Jauvtis and Williamson [22]. In order to simulate the physical initial condition of the circulating water tank, a still water was adopted in the whole computational domain.

**Table 1.** The structure and fluid parameters.

| Parameters | Value | Unit |
|---|---|---|
| Diameter of the cylinder D | 0.0381 | m |
| Mass ratio $m^*$ | 2.6 | - |
| Spring stiffness K | 25.86 | N/m |
| Damping ratio $\zeta$ | 0.00381 | - |
| Natural frequency in water $f_{nw}$ | 0.4 | Hz |
| Fluid density $\rho$ | 995.3 | kg/m$^3$ |
| Kinematic viscosity $v$ | $7.877 \times 10^{-6}$ | m$^2$/s |

The reduced velocity $U^*$ is defined as $U^* = U/f_{nw}D$, where $U$ is the incoming velocity; $f_{nw}$ is the natural frequency of the cylinder in still water; D is the diameter of the cylinder. The frequency ratio $f_y^*$ and $f_x^*$ are defined as $f_y^* = f_y/f_{nw}$ and $f_x^* = f_x/f_{nw}$, where $f_y$ and $f_x$ are the transverse and streamwise vibration frequency, respectively, taken from the FFT of the time history of the displacement.

Mesh independent study was conducted on four different mesh systems to check the spatial convergence of the numerical model. An example of flow past an elastically mounted circular cylinder at $U^* = 9$ was considered. The response amplitude in the transverse direction was sampled and compared with the published literature. The relative error between the present results and experimental results are defined by Equation (21) and the results are shown in Table 2. It should be noted that the mesh quality of Mesh 4 is higher than Mesh 3 even though the same nodes is employed on the circumference. The numerical results agree well with the experimental data as the mesh density increases. The relative error for Mesh 4 is found to be 4.2%, which is sufficient to obtain accurate results. The resolution of the grids on circumference of the cylinder with surface bulges is consistent with Mesh 4.

$$\varepsilon = \left| (y/D)_n - (y/D)_{exp.} \right| / (y/D)_{exp.} \qquad (21)$$

**Table 2.** Mesh independence study.

| Mesh | Node on the Circumference | $y$/D | $\mathcal{E}$ |
|---|---|---|---|
| Mesh 1 | 160 | 0.542 | 26.5% |
| Mesh 2 | 240 | 0.595 | 19.3% |
| Mesh 3 | 320 | 0.698 | 5.3% |
| Mesh 4 | 320 | 0.706 | 4.2% |
| Mesh 5 | 400 | 0.672 | 8.8% |
| Exp. [20] | - | 0.737 | - |

*3.5. Numerical Method Validation*

Based on the experiment of Jauvtis and Williamson [22], the VIV of a smooth cylinder, i.e., *CR*0 was simulated to validate the accuracy of the numerical method employed in this paper, and the results were compared with the published literature [24,25]. The results obtained in a smooth cylinder case will be used as the baseline of a cylinder with different coverage ratios. The reduced velocity investigated in this paper is over the range $U^* = 2 \sim 13$ with an increment of $\Delta U^* = 1$.

Figures 6 and 7 exhibit the variation of response amplitude in the streamwise and transverse direction respective. It can be clearly seen that the numerical model used in the present study successfully captures the VIV characteristics of a cylinder with a small mass ratio $m^*$, such as Streamwise Symmetric branch (SS) and Streamwise Antisymmetric branch (AS) and Initial branch (I), Super Upper Branch (SU) and Lower branch (L). Figures 6 and 7 show good agreement between the experimental and numerical data. There is a well-defined lock-in regime, where the transverse amplitude reaches its peak value $y/D = 1.51$

at the reduced velocity $U^* = 8$. During the calculation, it is found that the SU branch cannot be captured if a hybrid initialization is assigned. In the physical experiment, the flow velocity starts from still to a pre-set value. Some disturbances generate at the acceleration stage, which affects response amplitude of the cylinder.

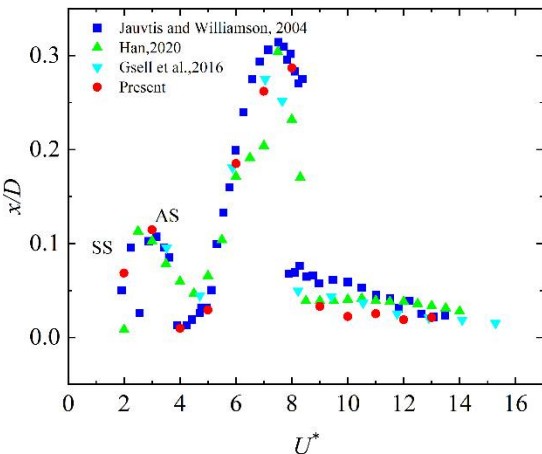

**Figure 6.** Variation of streamwise amplitude with reduced velocity [22,24,25].

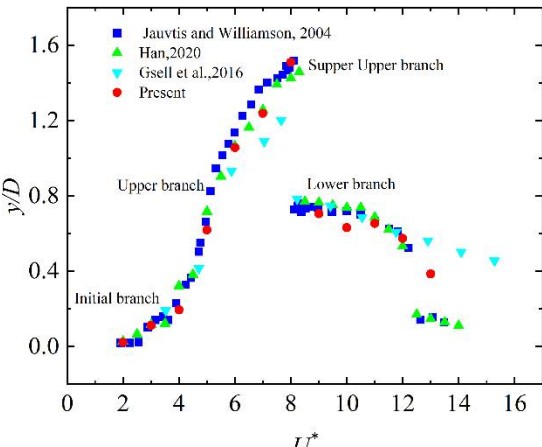

**Figure 7.** Variation of transverse amplitude with reduced velocity [22,24,25].

Figure 8 shows six typical VIV trajectories at selected reduced velocities. For $U^* = 2$, the amplitude of the streamwise vibration is larger than that of the transverse vibration. The vibration frequency in the streamwise direction is close to the natural frequency and a nearly flat Figure 8 shape is formed. With an increase of reduced velocity to $U^* = 3$, the VIV trajectory becomes more regular and shows the traditional Figure 8 shape. For $U^* = 4$, the vibration of the cylinder is in the transition stage from the I branch to SU branch, and the trajectory becomes chaos again. The vortex shedding frequency is highly affected by the natural frequency, and the transverse vibration exhibits the beat phenomenon (See Figure 9). For $U^* = 6 \sim 13$, the trajectory of the cylinder shows an obvious Figure 8 shape or crescent. In I branch and L branch stage, the trajectory is a Figure 8 shape; in SU branch stage, the trajectory is a crescent shape.

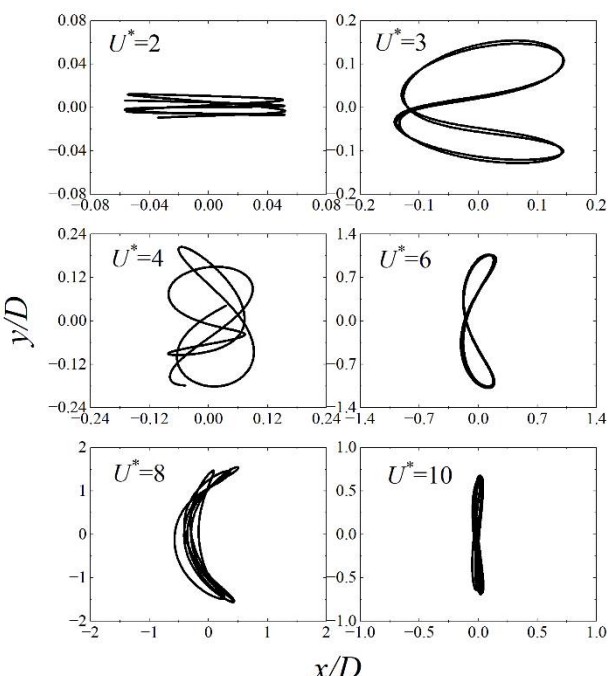

**Figure 8.** Typical VIV trajectories at selected reduce velocities.

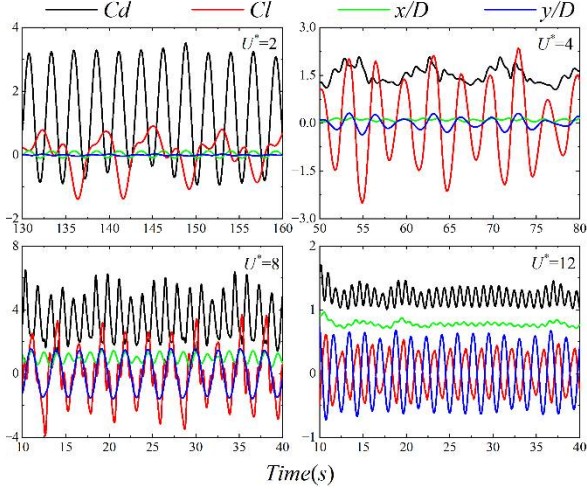

**Figure 9.** Time history of the force coefficients and displacement for smooth cylinder.

Figure 9 is the time history lift coefficient *Cl*, drag coefficient *Cd*, displacement in streamwise direction $x/D$ and transverse direction $y/D$ at selected reduced velocities. For $U^* = 2$, the streamwise vibration is locked, the drag coefficient is relatively large and the lift coefficient is chaotic. For $U^* = 4$, the transverse vibration exhibits the beat phenomenon. For $U^* = 8$, the vibration is in the lock-in regime and the response amplitude of the cylinder in both streamwise and transverse direction increases. For $U^* = 12$, the vibration enters into the L branch. The force coefficients, and response amplitudes decrease. There is a phase switch between the lift and transverse displacement from $0^\circ$ to $180^\circ$.

Variation of the transverse vibration frequency with reduced velocities is shown in Figure 10. The Strouhal frequency is also included for comparison. The vibration frequency is consistent with those published literature. There are three distinct frequency branches, which corresponds to the SS and AS branch, I and SU branch, and L branch, respectively.

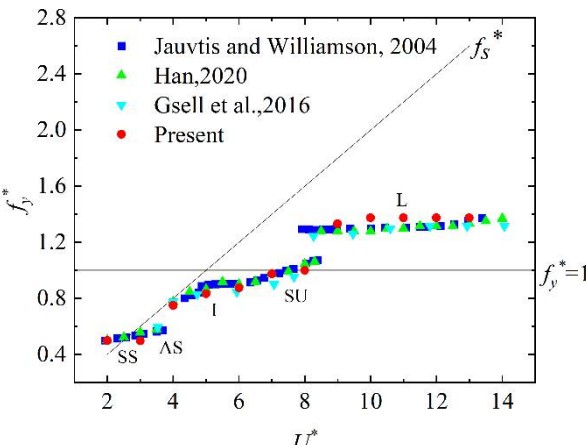

**Figure 10.** Transverse vibration frequency versus reduced velocity [22,24,25].

It should be noted that the time history of the transverse displacement exhibits a multiple frequency phenomenon in the transition regime and only the primary frequency is shown herein. In the first frequency branch, the ratio of the vibration frequency to natural frequency is $f_y^* = 0.5$. In the second frequency branch, the vibration frequency increases monotonously with the reduced velocity. For $U^* = 4 \sim 5$, the vibration frequency is approximately to that of $St$. For $U^* = 6 \sim 8$, the ratio of the vibration frequency to the natural frequency is close to $f_y^* = 1.0$, and vibration of the cylinder is located in the lock-in regime. In the third frequency branch, the ratio of vibration frequency to natural frequency is approximately $f_y^* = 1.4$. Meanwhile, the dimensionless vibration frequency of the streamwise direction is always twice the transverse direction, as shown in Figure 11. For the streamwise vibration, the vortex shedding on both sides of the cylinder causes once vibration, respectively, while an alternate vortex shedding completes one vibration period for the transverse vibration.

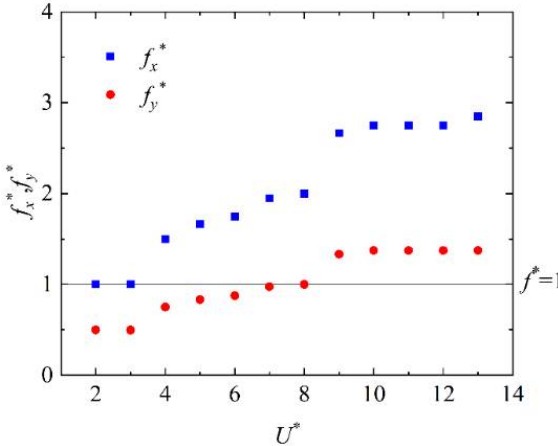

**Figure 11.** Vibration frequency versus reduced velocity.

Figure 12 depicts four typical vortex shedding modes for a cylinder with low mass ratio. For $U^* = 2$, the response amplitude is larger in the streamwise direction but smaller in the transverse direction as mentioned above, the wake mode is denoted as SS mode. For $U^* = 3 \sim 6$, the wake mode exhibits a traditional 2S mode. For $U^* = 7 \sim 8$, the transverse amplitude increases dramatically owing to the formation of a stronger triple vortex pairs, i.e., 2T mode. The vibration of the cylinder is located in the L branch for $U^* > 9$, and the wake mode depicts as 2P mode. With further increase of the reduced velocity ($U^* \geq 13$), the transverse response amplitude decreases dramatically and maintains at a small level. The 2S mode reappears in this desynchronization zone.

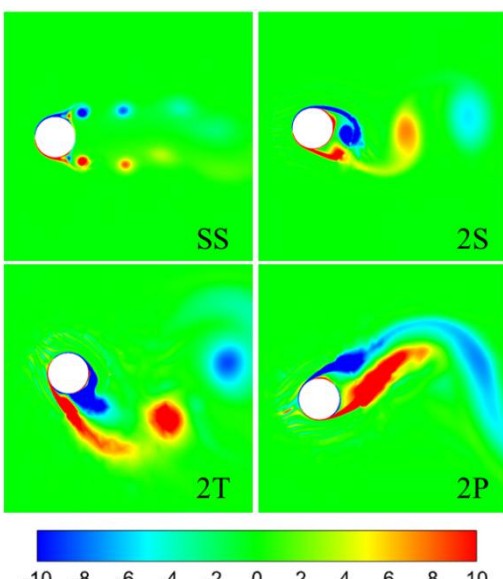

**Figure 12.** Four typical vortex shedding modes.

## 4. Results and Discussion

In this section, the VIV of a cylinder attached with surface bulges was carried out. The influence of coverage ratio on the hydrodynamic characteristics of the cylinder was considered. The parameters of the cylinder and fluid flow are taken as those of smooth cylinder. The performance of surface bulges on VIV suppression and the physical mechanism behind it were analyzed from the perspective of forces, vibration amplitude and frequency, trajectory and vorticity contour.

### 4.1. Vibration Amplitude

Figures 13 and 14 exhibits the variation of vibration amplitude of a cylinder with different coverage ratios with reduced velocity. It can be seen that the existence of surface bulges decreases the maximum vibration amplitude of the cylinder, and the reduced velocity corresponding to the peak value shifts from $U^* = 8$ to $U^* = 7$.

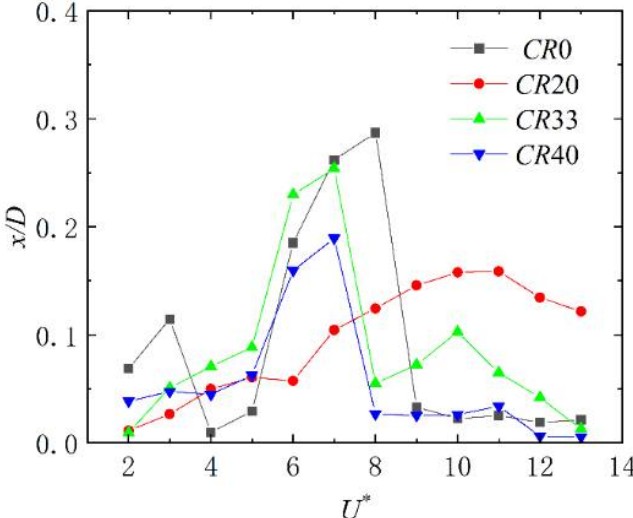

**Figure 13.** Vibration amplitude of a cylinder with surface bulges in streamwise direction.

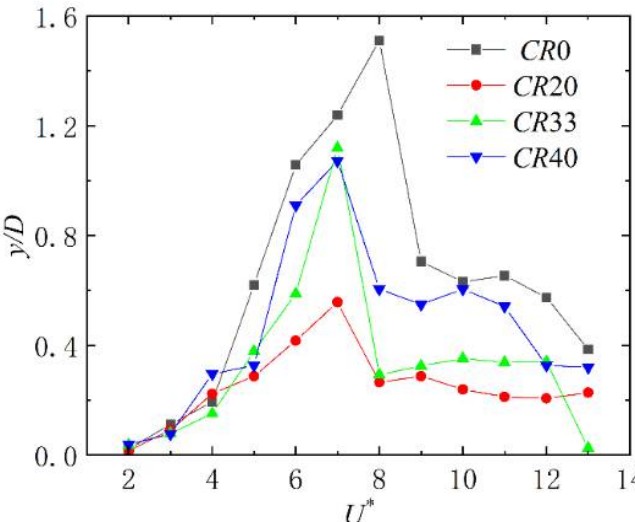

**Figure 14.** Vibration amplitude of a cylinder with surface bulges in transverse direction.

The maximum vibration amplitude of a cylinder with surface bulges at different coverage ratios were summarized in Table 3. The value in brackets is defined as the relative error of vibration amplitude between bulged and smooth cylinder. The values in the bracket are the efficiency of a cylinder with different coverage ratios. As can be seen from Table 3 that *CR*20 performs the best in VIV suppression. The suppression efficiency in streamwise and transverse direction are 44.6% and 63.1%, respectively. With the increase of coverage ratio, the VIV suppression efficiency gradually weakens. *CR*33 and *CR*40 are well-matched in the transverse VIV suppression. The variation trend of *CR*40 is analogous to that of smooth cylinder, which indicates that the VIV suppression capacity of the surface bulges reaches its limitation, and further increase the coverage ratio will not achieve a better performance in VIV suppression.

**Table 3.** The maximum vibration amplitude of a cylinder with surface bulges at different coverage ratios.

| CR | $x_{max}$/D | $y_{max}$/D |
|----|-------------|-------------|
| 0  | 0.287       | 1.511       |
| 20 | 0.159 (44.6%) | 0.557 (63.1%) |
| 33 | 0.254 (11.5%) | 1.120 (25.9%) |
| 40 | 0.189 (34.1%) | 1.072 (29.1%) |

*4.2. Motion Trajectory*

In Figure 15, with the increase of coverage ratio, the trajectories of a cylinder attached with surface bulges become regular and analogous to that of smooth cylinder. The trajectory of *CR*20 is chaotic over the reduced velocities investigated, and the traditional Figure 8 shape is no longer recognized. Both the vibration frequency in the streamwise and transverse direction are close to the natural frequency. Under the combined coupling effect of streamwise and transverse vibration, the vibration trajectory of the *CR*20 is chaotic. The trajectory of CR33 exhibits flat Figure 8, Figure 8, crescent and horizontal Figure 8. A horizontal Figure 8 shape trajectory is recognized for *CR*33 at $U^* = 13$, which was also reported by Zhou et al. [26]. According to Figures 16–18, the force coefficient and vibration amplitude are fairly small, and the transverse amplitude is only 6.6% of the smooth cylinder (see Figure 14). It is interesting to find that vortex shedding frequency in streamwise direction is close to that in transverse direction, giving rise to a horizontal Figure 8 trajectory. As the coverage ratio increases to *CR*40, the trajectory of the cylinder become regular, which is analogous to the smooth cylinder. However, the magnitude of the streamwise amplitude is 56% of the smooth cylinder at $U^* = 2$, thus the flat Figure 8 is not available.

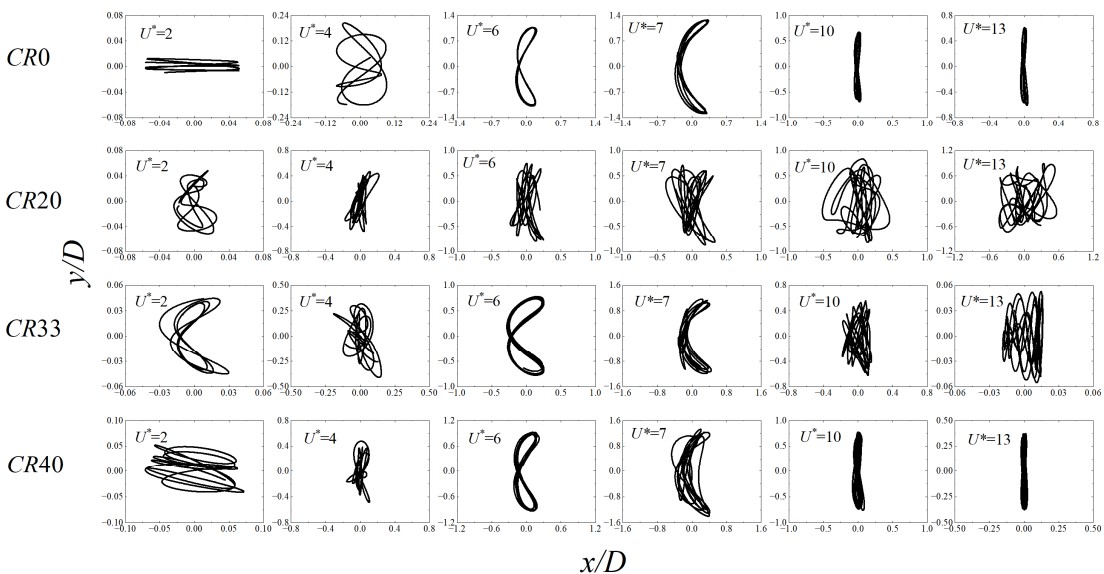

**Figure 15.** Motion trajectories for a cylinder with different coverage ratios.

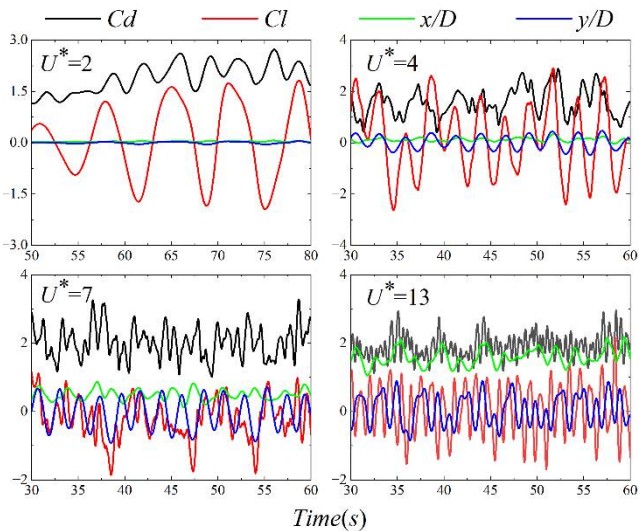

**Figure 16.** Time history of the force coefficients and displacement for *CR*20.

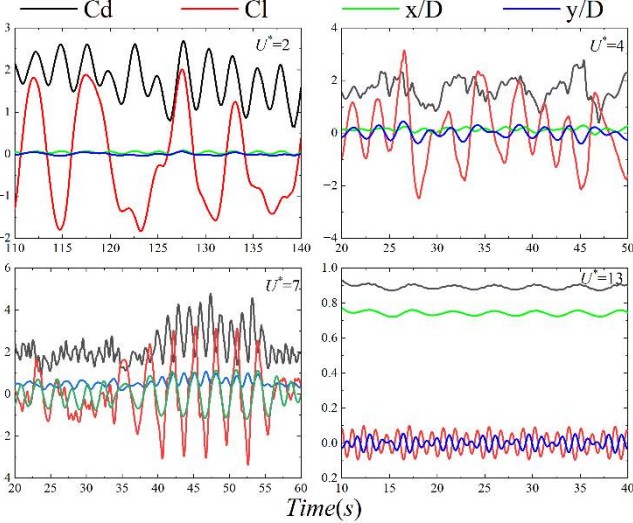

**Figure 17.** Time history of the force coefficients and displacement for *CR*33.

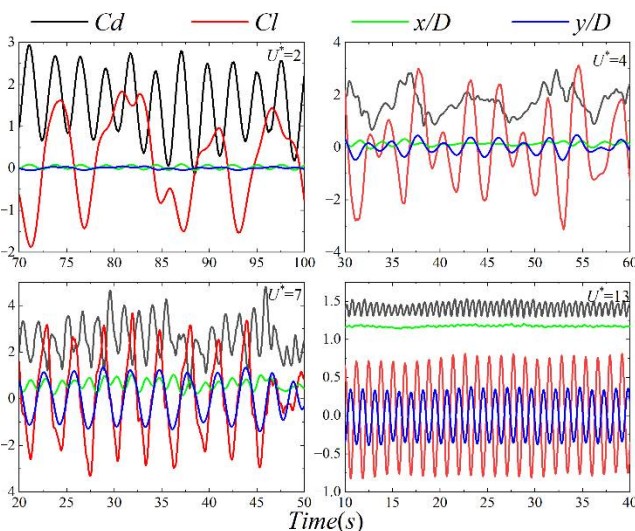

**Figure 18.** Time history of the force coefficients and displacement for *CR*40.

### 4.3. Force and Displacement of Structure

With the increase of coverage ratio, the forces and displacements of the cylinder with surface bulges gradually change from chaos to regularity. The streamwise vibration is in lock-in regime at $U^* = 2$ for *CR*20, *CR*33, and *CR*40 (See Figures 16–18). With the increase of coverage ratio, the drag coefficient of the bulged cylinder increases, and the lift coefficient becomes chaotic; the response amplitude increases in the streamwise direction gradually. The time history of the displacement for the bulged cylinder becomes chaotic at $U^* = 4$. The transverse displacement and lift coefficient exhibit multi-frequency phenomenon and gradually exhibit the beat phenomenon. The transverse vibration is in lock-in regime at $U^* = 7$ for the bulged cylinder (See Figures 16–18) and the magnitude of the vibration increases. With the increase of coverage ratio, the time history of transverse displacement and lift coefficient switch from chaos to regularity, suggesting the vibration frequency and lift frequency switch from multi-frequency to a single primary frequency. The magnitude of the force and displacement for all cylinders with surface bulges decreases at $U^* = 13$. There is a phase switch between the transverse displacement and lift coefficient from $0°$ to $180°$.

### 4.4. Vibration Frequency

In Figure 19, the vibration frequency of the bulged cylinder with different coverage ratios is shown. When the coverage ratio is low, i.e., *CR*20, the frequency is concentrated upon the natural frequency. The lock-in regime of the stream-wise and transverse vibration are over the range $U^* = 2 \sim 10$ and $U^* = 3 \sim 8$, respectively. Under the combined coupling effect of stream-wise and transverse vibration, it vibrates with a small magnitude and exhibits a chaotic state. With the increase of coverage ratio, the lock-in regime of the cylinder with surface bulges shrinks, and the trend of transverse vibration frequency follows that of a smooth cylinder. The lock-in regime of the transverse vibration is over the range $U^* = 4 \sim 7$ *CR*40 for and the transverse vibration amplitude increases sharply.

It is interesting to find that the transverse vibration frequency is larger than the stream-wise vibration frequency for *CR*20 and *CR*33 at large reduced velocities. Medium-size vortex forms due to the existence of the surface bulges, which alters the vortex shedding frequency and thus the fluctuating forces acting on the cylinder. With the increase of coverage ratio, the size of the vortex forms between the surface bulges decreases, which minimizes its influence on the vortex shedding. When the coverage ratio reaches 40%, the vibration frequency of *CR*40 is similar to that of smooth cylinder, and the vibration frequency in the streamwise direction is twice the transverse vibration frequency.

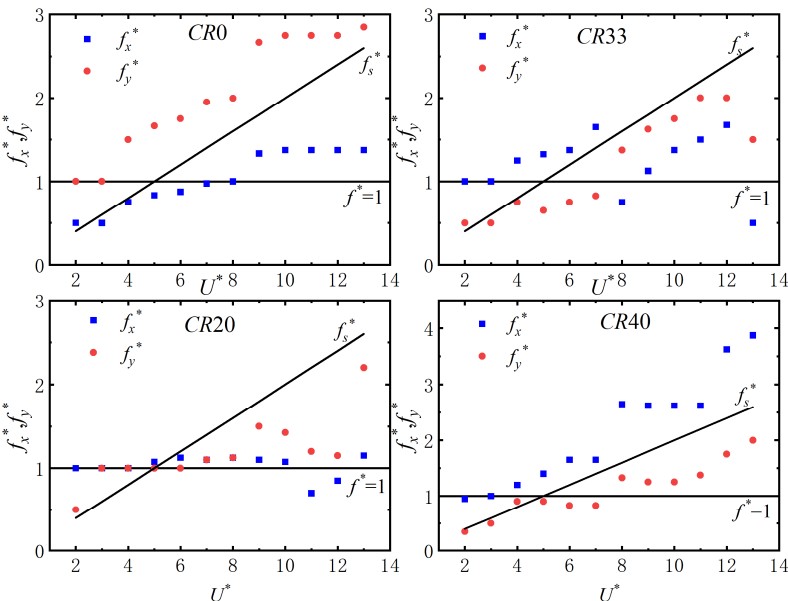

**Figure 19.** Vibration frequency of bulged cylinder with different coverage ratios.

### 4.5. Streamline and Vorticity

It can be seen from Figure 20, the main contribution of surface bulges on the VIV suppression of a cylinder is that those surface bulge affects the separation of the shear layer, and the movement of the fluid deflects. With the increase of coverage ratio, large-scale vortices in the vicinity of the cylinder split into small-scale vortices, which affects the induced vibration of the cylinder. In the case of the smooth cylinder, the separation of the shear layer is fixed at Point 1, and the separation point of a cylinder with surface bulges moves forward. It can be seen that there is a small vortex formed right behind the bugle, which causes an unstable VIV response and diminishes the vibration amplitude of the cylinder. As the coverage ratio reaches 40%, the bulges are relatively dense, leading to the flow pattern of *CR40* resembling that of a smooth cylinder. Several small vortices form in the area of Point 3 and 4, which weakens the VIV suppression efficiency of surface bulges.

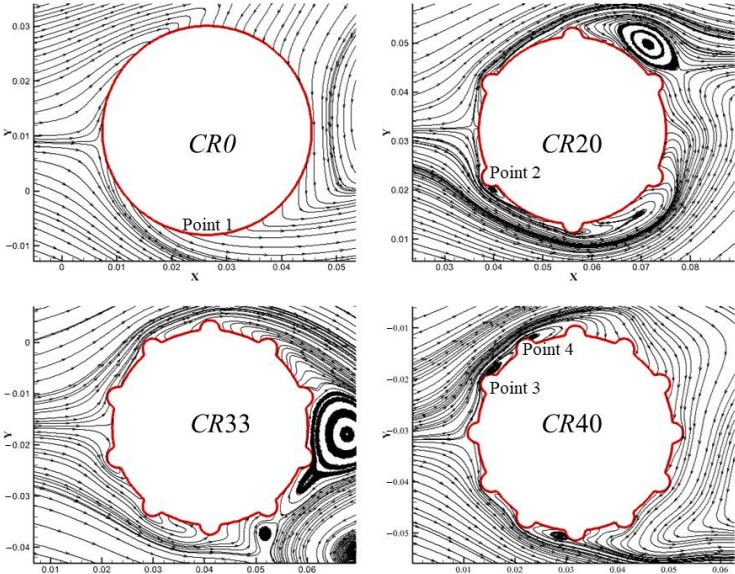

**Figure 20.** Close-up of the streamlines of a cylinder with different coverage ratios at $U^* = 11$.

Figure 21 is the vorticity contours for a cylinder with different coverage ratios at selected reduced velocities. In all frames, the cylinder is located at its maximum value.

With the increase of coverage ratio, the wake structures and wake mode become regular, which is resemble to that of smooth cylinder. For a cylinder with a lower coverage ratio, the bulge affects the separation of the shear layer, and the vortex shedding mode cannot be recognized. The distance between the adjacent vortex is relatively small and the vibration of the structure is greatly affected by the near wall vortex. With the increase of coverage ratio, the separation point of the shear layer moves forward, which affects the vibration characteristics of the structure. When the coverage ratio reaches 40%, the shear layer separates at the top of the bulge, weakening the vibration suppression effect of the bulge, and small vortices are formed between the two bulges, which makes a small contribution to the VIV suppression. A coverage ratio of 40% is dense enough, and the bulged cylinder can be simplified as a smooth circular cylinder with a diameter of 1.1 D, thus the hydrodynamic characteristics is similar to those of a smooth cylinder.

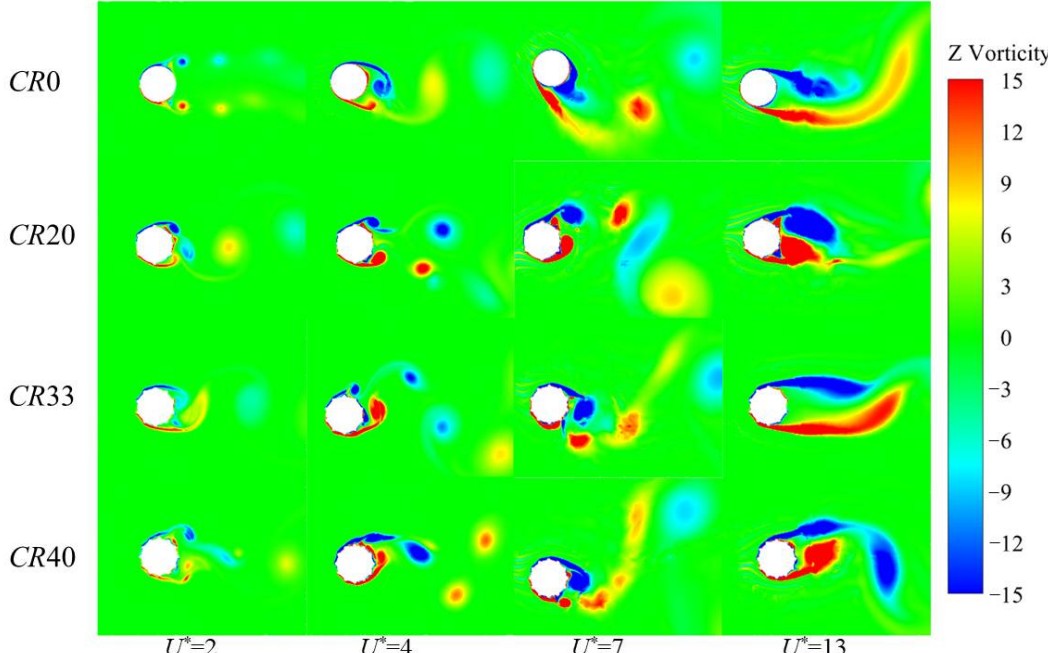

**Figure 21.** Vorticity contours for a cylinder with different coverage ratios at selected reduced velocities.

## 5. Conclusions

In this paper, the VIV of elastically mounted cylinder was realized based on the ANSYS Fluent and User Defined Function (UDF). The dynamic motion of the cylinder was solved by the single-step time integration algorithms Newmark-$\beta$ method. The in-house code was first validated by studying the 2DOF VIV of a circular cylinder with small mass ratio over the range $U^* = 2 \sim 13$, and the results agrees well with the published literature. Then, the performance of surface bulge on VIV suppression was studied and four different coverage ratios were considered, i.e., 0%, 20%, 33%, and 40%. The hydrodynamic characteristics of the bulged cylinder was investigated by the force coefficients, vibration amplitude and frequency, motion trajectory, and vortex shedding mode. The mechanism of surface bulge on the VIV suppression was investigated. The following are some major conclusions:

(1) The in-house code successfully captures the hydrodynamic characteristics of a cylinder with low mass ratio experiencing 2DOF VIV, including all response branches, i.e., the SS and SA branches in the streamwise direction and I, SU and L branch in the transverse direction, well-defined lock-in regime, wake modes, i.e., SS, 2S, 2T, 2P. The maximum transverse vibration amplitude reaches $y/D = 1.51$, which is consistent with the experimental results;

(2) The VIV of a cylinder attached with surface bulges can be effectively suppressed. *CR*20 performs the best in VIV suppression and the suppression efficiency in streamwise

and transverse direction are 44.6% and 63.1%, respectively. Moreover, the variation trend of *CR*40 is analogous to that of a smooth cylinder, which indicates that the VIV suppression capacity of surface bulges reaches its limitation, and a further increase in the coverage ratio will not achieve a better performance in VIV suppression;

(3) The mechanism of a surface bulge on the VIV suppression is the shift of the separation point of the shear layer and vortices form between the surface bulges. For a cylinder with a low coverage ratio, the distance between the adjacent vortex is relatively small and a medium-size vortex forms between the surface bulges. The vibration of the structure is greatly affected by the near wall vortex. With the increase of coverage ratio, the separation point of the shear layer moves forward and the size of the vortex forms between the surface bulges decreases, which minimizes its influence on the vortex shedding. When the coverage ratio reaches 40%, the surface bulges are dense enough and the bulged cylinder can be simplified as a smooth circular cylinder with a diameter of 1.1 D, thus the hydrodynamic characteristics is similar to those of a smooth cylinder.

**Author Contributions:** Conceptualization, H.X. and B.Z.; methodology, B.Z. and J.W.; formal analysis, Z.L. and K.L.; validation, B.Z.; writing—review and editing, H.X., J.W. and J.Y. All authors have read and agreed to the published version of the manuscript.

**Funding:** This research was funded by the National Natural Science Foundation of China (Grant Nos. 52071059, 52192692, and 52061135107); the LiaoNing Revitalization Talents Program (Grant Nos. XLYC1807190 and XLYC1908027); the Dalian Innovation Research Team in Key Areas (No. 2020RT03); the Fundamental Research Funds for the Central Universities (No. DUT20TD108); and the State Key Laboratory of Ocean Engineering (Shanghai Jiao Tong University) (Grant No. GKZD 010081).

**Institutional Review Board Statement:** Not applicable.

**Informed Consent Statement:** Not applicable.

**Data Availability Statement:** Some or all data, models, or codes generated or used during this study are available from the corresponding author upon request.

**Conflicts of Interest:** The authors declare no conflict of interest.

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
