# Peer review of "A Study on the Vortex Induced Vibration of a Cylindrical Structure with Surface Bulges"

_jmse, doi:10.3390/jmse10111785_

Round 1

Reviewer 1 Report

This paper needs further work to be comprehensible to anyone but its authors:

"VIV" must be spelled out in the title.  Vortex Induced Vibration?  Reader should not have to guess.

A reference for the Newmark beta method should be given.

What are the elastic constants and dimensions of the cylinder?

Minor comments:

Numerous minor grammatical and usage errors need correction by someone fluent in English.

l. 34: from, not form.

l. 81: mass ratios: ratios of which masses?

l. 174: Define turbulent intensity

Table 1: Again, mass ratio is ratio of which masses.  Natural frequency: which mode?

Figs. 8, 15: What are the abscissa and ordinate variables?

Table 3: What are the percentages.

Author Response

Dear Reviewers,

We appreciate the time the reviewers have dedicated to providing valuable feedback. In the following pages are our point-by-point responses (in blue font) to each of the comments (in Regular black font) of the reviewers as well as your own comments. Revisions in the manuscript are marked in red font for corrections. Based on the instructions provided in your letter, we have uploaded the revised manuscript onto the website. The responses to the reviewers’ comments are attached to this document for review and consideration.

Yours sincerely,

Zhou Bo (Prof)

Dalian University of Technology

Reviewer 2 Report

This paper deal with a highly studied problem. I make some recommendations:

Include more references. There are many research that can be included in your intruduction as "wake reduction" in bluff bodies. I add two:

https://www.sciencedirect.com/science/article/pii/S0889974618307825

https://www.sciencedirect.com/science/article/pii/S0889974618308065

Line 101: Navier Stokes.

line 127 to 129: rewrite. It is confusing.

Explain why in figure 3 you use _ for "Define_CG_..." in case that it is necessary use italics even in the text.

Indicate the y*

I do not think the mesh convergence is properly done. An extra case should be done (19.3 and 26.5 % error do not show that you are converging) 

What about time step convergence?

line 208 use between instead of with

from line 211 to 213 rewrite

line 311 C30? Should it be C33 ?

line 368 You use a different text format.

Figure 20. For a better comparison you should include axis in the plot. Besides it should be interesting to see more area of streamlines? Are they an average? 

Figure 21. Please use the same scale for both parts (one goes from -10 to 10 and the other from -20 to 20) 

Author Response

(The authors gave the same response as above.)
